# Evaluation of Gender-Related Digital Violence Training in Catalonia

Catalina Guerrero-Sanchez [1,*], Jordi Bonet-Marti [1,*] and Barbara Biglia [2]

1   Departament de Sociologia, Facultat d'Economia i Empresa, Universitat de Barcelona (UB),
    08034 Barcelona, Spain
2   Departament de Pedagogia, Facultat de Ciències de l'Educació i Pedagogia, Universitat Rovira i Virgili,
    43007 Tarragona, Spain; barbara.biglia@urv.cat
*   Correspondence: cguerrsa7@alumnes.ub.edu (C.G.-S.); jordi.bonet@ub.edu (J.B.-M.)

**Abstract:** This study examines the results of evaluating a Catalan training program for practitioners working with survivors of gender-related violence. Considering the lack of scientific evidence previously shown by studies on this topic, this article aimed to triangulate the participants' self-perception with their assessment of knowledge and competencies in tackling digital gender-related violence before and after the training. To do so, a pre-test and post-test case-based design was employed to identify and measure the participants' improvement in self-perceived knowledge and their effective gain in knowledge and skills to address this kind of violence. Considering the contributions of a feminist evaluation approach, we also included in our evaluation the analysis of classroom interactions and the participants' responses. The results overall demonstrate that the incorporation of assessment criteria from the feminist evaluation methodology increased the reliability of evaluation criteria. In addition, it also enabled us to identify the need to continue developing training programs that empower participants and prevent women and LGBTQI+ people from disengaging from digital spaces.

**Keywords:** evaluation; gender-related violence; violence; training; feminism

## 1. Introduction

Today's growing digitalization has led to significant benefits in the daily lives of many individuals, making it easier for some to communicate, work, and learn. However, digitalization has also created a new space in which gender-related violence, understood as violence against women and girls and homophobic and transphobic violence based on gender inequality and gender order (Alldred and Biglia 2015; Biglia and Martín 2007) persists and intensifies. According to Vergés and Gil-Juarez (2021), women and LGBTIQ+ people frequently face violence in digital spaces, and they express more fear than heterosexual men of being targeted online. One of the key debates that stands out in the literature concerns the dual effect that digital spaces have on gender-related violence. On one hand, they have served as a significant platform for mobilization, particularly by feminist and LGBTIQ+ movements, while, on the other hand, they have offered new opportunities for the amplification of gender-related violence (Dragiewicz et al. 2018; Sills et al. 2016). In this context, Simões et al. (2021) suggest that the same characteristics of the digital environment that have empowered the feminist movement have also enabled the amplification of digital gender-related violence.

International organizations warn us about the impacts of this type of violence, which range from harm to a person's reputation through psychological disorders to self-injury and suicide in extreme circumstances (Khoo 2021). Previous studies have found that survivors of this type of violence are more likely to be absent from schools and workplaces, resign from their jobs or studies, and avoid social events (Hill and Johnson 2019). Additionally, the most common response by women and LGBTQI+ in order to face this expression of violence

is to reduce their online activities by disconnecting from social networks or self-censoring; in this way, gender-related digital violence (GRDV) is also impacting women's political participation, freedom of expression, and mobilization (Khoo 2021).

Besides this, the COVID-19 pandemic has revealed the necessity to develop innovations in responding to GRDV (Jewkes and Dartnall 2019). Thus, after the proliferation of GRDV during the lockdown, the United Nations Population Fund recommended strengthening survivor-center response services through continuous training and capacity building for healthcare and psychosocial service providers and social workers, as well as other service providers (UNFPA 2021).

The existing literature also points out that technological tools have provided new opportunities for "social movement learning" (Hall and Clover 2005). In this context, social media platforms offer a way to increase the scale of participation in social movements, allowing diverse audiences to engage not only in disseminating messages, but also in political and pedagogical practices (Simões et al. 2021). Among these practices, we must highlight the training intended for activists and professionals to counter digital violence against activist and minority groups.

According to the literature, in recent years, we have witnessed an increase in intervention programs and policies for preventing gender-related violence. However, there is still limited evidence to confirm the effectiveness of these interventions or training programs (Seff et al. 2021; Sharma et al. 2022). The growing number of trainings has also led to consider the effectiveness of evaluation models. Musungu et al. (2018), among others, suggest that online training is a viable option, especially for the training of professionals. Besides this need for more rigorous evaluations, certain key factors necessary for designing and conducting online workshops have also been highlighted.

A compilation of studies on the evaluation of programs for gender-related violence prevention indicates that, over the past 25 years, we have witnessed progress in the formulation of universal prevention strategies to reduce gender-related violence, especially in schools and university campuses (Crooks et al. 2019). However, the authors acknowledge the lack of evidence regarding how to prevent violence in settings outside school contexts, encouraging students to make this the focus of future research (Crooks et al. 2019). Besides this, most training evaluations are only based on self-perceived improvement and do not incorporate other elements that should be considered (Etherington et al. 2017). Furthermore, there has been limited progress in the field of GRDV training.

In line with this, a prior analysis of evaluation reports on projects related to violence against women (2008–2012) indicates that 69% of the selected evaluations used a primarily qualitative approach (Dragiewicz et al. 2018). Moreover, the authors highlight the limited scientific evidence provided by many of these evaluations, as they gather evidence solely from an interested group—mostly from the implementing team—without triangulating their perspective with that of the supposed beneficiaries or other stakeholders.

As several articles point out, this gap in the field of program evaluations may well be related to the limited budget that is sometimes allocated to the evaluation in projects addressing gender-related violence (Biglia et al. 2022; Crooks et al. 2019; Donoso Vázquez 2012; Etherington et al. 2017). Some authors suggest that this lack of indicators and measurement of the effects of training in gender-related violence can create an illusion of progress in addressing this issue when, in fact, the available resources and opportunities for comprehensive educational processes may not be being used to their full potential.

Regarding the methodological design employed for evaluating interventions in the field of gender-related violence, one of the most common options in the literature is randomized controlled trials (Crooks et al. 2019; Halim et al. 2019; Jewkes et al. 2019; Ogum Alangea et al. 2020). However, Crooks et al. (2019) point out that while this research design has shown multiple strengths in previous studies, it may not always be considered the best option, as its success is not always transferable to real-world scenarios.

Raab and Stuppert (2018) present the results of a comparative study of existing evaluations in the field to identify the conditions leading to effective and high-quality evaluations.

They understand such kinds of evaluations as the ones that strengthen the course of an intervention, influence policymakers, attract donors, or increase participant learning. According to the authors, the evaluations that achieved these desired effects were developed considering four key elements: (1) they present evidence with robust data and transparent documentation; (2) they are gender-sensitive; (3) they maintain active consultation with key stakeholders; and (4) they effectively communicate the results (Raab and Stuppert 2018).

Regarding the first element, data collection is one of the difficulties highlighted in the literature concerning the evaluation of intervention programs in gender-related violence. This includes issues related to the security and the ethics of data collection, as well as the accurate identification of the type of data needed to determine the success of an intervention (Sharma et al. 2022). The danger of misinterpreting the results of an evaluation has also been emphasized since it can lead to overlooking the complexity of gender inequality realities. For instance, an increase in reports of gender-related violence after an intervention does not necessarily indicate a rise in violence but rather increased awareness, identification, and/or reporting of such incidents (Dragiewicz et al. 2018).

This exemplifies the importance of considering gender sensitivity as a crosscutting element of both training and evaluation. Gender sensitivity can be materialized in the evaluating team's familiarity with gender inequality research and the management of risks when investigating these forms of violence (Raab and Stuppert 2018). This gender sensitivity becomes even more relevant when we recall that intervention programs designed to address gender-related violence have begun to adopt a gender-neutral approach (Pagani et al. 2022). In this context, it is worth remembering some of the contributions made by the evaluation frameworks adopted from a feminist perspective, where the emphasis is placed on the process rather than just the outcomes (Biglia et al. 2022).

Observing the process is particularly relevant considering that the way training programs are implemented can have a direct impact on the program's outcomes. For instance, a study evaluating a gender-related violence intervention program (Mentors in Violence Prevention) noted that variations in implementation could partly explain the conflict in the program's results (Pagani et al. 2022).

To close this section, we want to highlight three key ideas. Firstly, the lack of rigorous evaluations despite the increasing number of gender-related violence training programs. Secondly, the absence of evaluations for training programs specifically targeted at professionals working with survivors of gender-related violence. Finally, the need to develop gender-sensitive evaluations of the training outcomes and the implementation process. All these gaps in the literature provided the base for the research objectives and questions in this work.

Given that the evaluation is a crucial element for the enhancement and monitoring of GDRV training, the following study examines the program's outcomes under consideration by incorporating aspects of feminist evaluation. Our study was guided by the following research question: How can we improve GRDV-training evaluation schemes from a feminist perspective to assess knowledge and skill acquisition?

## 2. Methodology

This article presents the evaluation design and results of the training delivered to practitioners working with women survivors of gender-related digital violence (GRDV). The courses aimed at raising the participants' awareness and equipping them with tools for mitigating GRDV. They were developed in the framework of the FemBloc Project, which is an action-research initiative funded by the European Commission to provide tools and training to tackle GRDV in Catalonia (Cruells et al. 2021).

The courses targeted professionals in four different Catalan women's services. Their duration varied according to the needs of the involved institutions, ranging from a single session to up to four. In total, eighteen online sessions, divided into three blocks, were carried out. The first block was on the legal framework of gender-related digital violence, and its key concepts and its distinct categories were introduced. The second was devoted

to the impact of GRDV, its prevention strategies, and the responses to it. Finally, in the third block, tools and strategies to counter GRDV were presented by studying particular cases. The training program also included three activities, namely, a test on digital security, the installation of digital tools for the prevention of GRDV and the protection against it, and the analysis of a practical instance.

The research methodology employed incorporates different elements of feminist evaluation methods rising from the proposal by Biglia et al. (2022) concerning the need to consider aspects beyond the parameters of efficiency and effectiveness that commonly guide evaluations. In this sense, feminist evaluation is proposed as an option for monitoring interventions aimed to produce social transformation and gender sensitivity. The feminist evaluation conception of assessment as a continuous learning process rather than an exercise of examination or control is also particularly important since evaluation has commonly been associated with negative connotations. For example, evaluations can have punitive consequences, such as participant withdrawal from a specific program, even when the interpretation of data might not be accurate, as explained earlier (Donoso Vázquez 2012; Dragiewicz et al. 2018). From this perspective, to consider both the process of training as well as its results, Biglia et al. (2022) propose six evaluation criteria for monitoring training on gender-related violence: positionality, interaction, care, response, influence, and diffraction. The empirical design of this research has adapted this proposal as presented in Table 1, where we identify the elements to be tracked in accordance with each of the criteria outlined. Unfortunately, we could not analyze the diffraction, as it involves extensive long-term monitoring and aspects such as positionality, care, and influence were approached with a shallower level of scrutiny due to time constraints.

**Table 1.** Feminist evaluation criteria.

| **1. Positionality** |
| --- |
| -To observe if the training design is suitable for the profile, knowledge, or motivations of the participant group, incorporating cases and examples relevant to their experiences.<br>-To observe the extent to which the understanding and addressing of gender-related violence is approached by the facilitators, recognizing its structural and systemic nature, and institutional violence, while addressing the intersectionality of its causes, meanings, and effects.<br>-To evaluate whether experiential knowledge is given a prominent place, offering concrete examples and cases. |
| **2. Classroom interactions** |
| -To identify whether the experiences, knowledge, and opinions of the participants are valued by the facilitators.<br>-To identify whether discussion of different perspectives or interpretations is encouraged, with an emphasis on respecting the experiences and emotions triggered during the process.<br>-To evaluate whether a space is provided for participants to question the provided content or interpretations. |
| **3. Care** |
| -To observe whether facilitators can identify and respond to difficulties, resistance, or discomfort from participants towards specific dynamics.<br>-To take into account whether schedules and calendars facilitate work–life balance, comfort, and accessibility to the training.<br>-To evaluate the working conditions of team members and the implementation of personal and collective self-care tools. |
| **4. Participants' response** |
| -To analyze participants' satisfaction with the proposed dynamics.<br>-To demonstrate how participants perceive the personal and/or professional usefulness of the training program.<br>-To indicate whether the participants show interest in finding out more about GRDV. |
| **5. Influence** |
| -To highlight participants' perception of their having internalized knowledge and skills. |

Note: The table was developed by the authors based on the feminist evaluation model by Biglia et al. (2022).

In addition to the criteria described above, contextual elements indicating the characteristics of the participants in the training sessions and the facilitators/instructors (Biglia et al. 2022) were taken into account.

*2.1. Research Techniques and Data Analysis*

A quasi-experimental research design was proposed to measure changes in participants' GRDV self-perceived knowledge. This type of design is especially employed when studying the impact of an intervention (Ato et al. 2013), and, in our case, it provided insights into the influence of the training program on the participants. This quasi-experimental design has previously proven to be useful in collecting quantitative data to support research on gender-related violence (Puigvert et al. 2019). The participants in the training sessions were intentionally selected, and no control group was implemented.

A first questionnaire was administered at the beginning of the sessions (pre-test), and a second one was conducted at the end of the training (post-test). The difference between the scores on knowledge and skills obtained after the training program and the scores at the beginning of the program allowed us to have a quantitative indicator of the perception of learning. These measurements, before and after the implementation of the training, have been mentioned as a potential best practice in the field of gender-related violence assessment (Sharma et al. 2022).

The T-test for independent samples was employed to assess whether there was a significant change ($p < 0.05$) between the means of pre-test and post-test scores. The null hypothesis in this case ($H0^1$) indicated that there would be no significant difference. In other words, if we confirm the null hypothesis, we shall conclude that there was no substantial change in self-perceived knowledge about GRDV after the training.

To evaluate the participants' skills acquisition, each questionnaire included a fictional case, drafted and validated by experts, in which the type of GRDV had to be identified and a possible response to the aggressions suggested. In each of them, participants were asked to identify three kinds of violence and list up to four actions they would take to respond to the proposed situation. The assessment of these responses was carried out with the assistance of an evaluation rubric by one of the facilitators along with a member of the evaluation team. A paired samples T-test was employed to analyze the results concerning skills. In this way, it was possible to cross-reference self-perceived knowledge from the participants' perspective with a measure taken from the perspective of the evaluation team.

Finally, a participant observation exercise was conducted during the virtual sessions to document the training process. The observation notes were later analyzed based on the previously described criteria of feminist evaluation (positionality, interactions, care, response, and influence). Additionally, the post-training questionnaire included some items in which participants rated the training program content, the teaching methods used, and their learning experience. Open-ended items were also included so that participants could share what they liked most about the training and areas for improvement.

The pre-test and post-test questionnaires were administered through remote data collection. According to some studies, online questionnaires help mitigate interviewer fatigue and thus promote the quality of the collected data in contrast to face-to-face questionnaires (Seff et al. 2021). However, this data collection technique involves additional ethical and methodological considerations, such as increased data vulnerability, lower participation rates, and retention throughout the research (Seff et al. 2021). In line with the required care for the participants and their data, their consent was obtained before completing the questionnaires.

Table 2 summarizes the data collection tools: the pre-test questionnaire, the post-test questionnaire, and the observation diary.

**Table 2.** Data collection tools.

| | | |
|---|---|---|
| Questionnaire—pre-test | Goals | -To measure the initial state of self-perception of knowledge and skills in relation to GRDV.<br>-To collect sociodemographic data of the participants, their motivations, expectations and experience (positionality criteria) |
| | Target audience | Healthcare professionals beginning the training sessions |
| | When it is applied | At the beginning of the first training session |
| | How it is answered | Individually, online |
| | Time | 15/20 min |
| Questionnaire—post-test | Goals | -To measure the final state of self-perceived knowledge and perceived skills in relation to GRDV from the point of view of the evaluation team<br>-To gather the participants' perceptions of interactions and care within the classroom, as well as their responses and the influence of the training |
| | Target audience | Professionals who have completed at least three training sessions |
| | When it is applied | At the end of the last training session |
| | How it is applied | Individually, online |
| Observation diary | Goals | -To collect elements of the training process such as the kind of classroom interactions during the training |
| | When it is applied | During five training sessions |

Note: Authors design based on Donoso Vázquez (2012).

## 2.2. Population, Sample, and the Implementation and Evaluation Teams

A total of 125 professionals participated in the training session, divided into seven groups. The survey response rate was 82%; thus, the sample for our quantitative evaluation was 104 individuals. Regarding the participants' sociodemographic information, 97.12% of them self-identified as women and 2.88% as non-binary. Their average age was 39.7 years; 78.65% were born in Catalonia, 13.48% in another region of Spain, and 7.87% in a non-EU country. Most participants had experience in attending women survivors of gender-related violence for more than one year (70.53%).

The facilitation team included experts in software development and techno-activism, a criminal lawyer specializing in technological and LGBTQ+ rights, and psychologists specializing in gender-related violence. All had a prior history in the field of feminist digital defense, and while most were from Catalonia or other cities in Spain, one was originally from Cuba.

The evaluation was carried out by the University of Barcelona team. Author 2, from Catalonia, has devoted part of his career to researching antifeminist discourses on social media, studying gender-related violence and feminist research methodologies. Author 1 is a naturalized Colombian citizen with experience in gender-focused research and programs aimed at reducing the gender digital divide in Latin America and providing support for survivors of gender-related violence. Finally, Author 3, an Italian scholar based in Catalonia for 30 years, is a specialist in feminist evaluation and gender-related violence and acted as a methodologic advisor.

## 3. Results

The results are presented in three sections: the description of training implementation; the analysis of the self-perceived knowledge acquisition; and the changes in competencies and effective knowledge gained, based on the assessment of practical cases.

Before that, considering the importance of understanding how much the training responds to the participant's motivation and expectations for enrolling, we present an overview of these elements.

As shown in Figure 1, most participants (80.9%) were interested in developing tools and skills to apply to their work, while 62.92% were motivated to receive training where theory and practice were integrated. On the other hand, 93.26% expressed that their motivation was not the result of an explicit request from their supervisors. The possibility of debating with others was not a major motivation for attending the course, and 98.88% did not expect to engage in debates with the rest of the group. Additionally, 91.01% of the involved professionals did not wish to create networks with other professionals. Instead, 91.01% expected to expand their knowledge about digital gender-related violence, and 73.03% desired to acquire tools for their work.

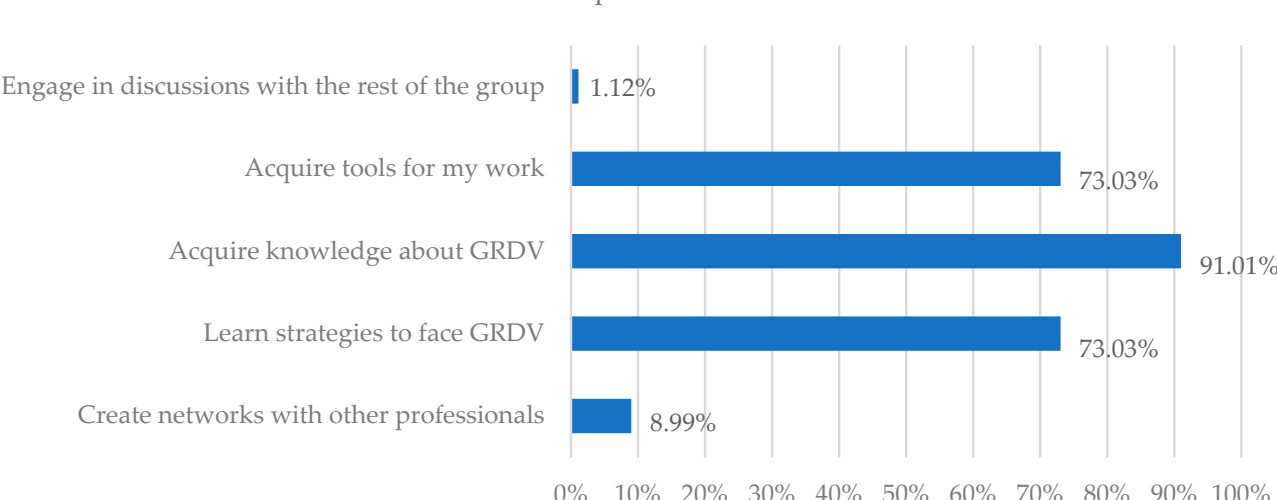

**Figure 1.** Motivations and expectations of the participants.

### 3.1. Training Implementation

The observation of the training session and some of the responses to the questionnaires allow us to describe and analyze the process of the implementation based on the five criteria of Table 1: positionality, classroom interaction, care, response, and influence.

Concerning positionality, during the session it was possible to observe how the facilitators conceptualized GRDV as a structural and systemic phenomenon. They devoted a significant part of the introduction to highlighting the economic interests of large tech companies in the selling of data. Furthermore, dynamic discussions were observed, where professionals were asked to share their experiences in addressing cases of GRDV in their institutions. Despite the low motivation and expectations to engage in discussions with their peers—as shown above—the participants highly rated the usefulness of the group dynamics with a 7.03 out of 10.

Regarding the interactions in the virtual classroom, the facilitators frequently encouraged participants to ask questions and share opinions. Consequently, frequently enough, participants shared experiences and examples from their services, and requested further exploration of specific topics (for example, how to achieve evidence verification in a legal process). However, as Figure 2 shows, when participants were asked about their impressions of the teaching methods, the "Balance between theoretical and practical content" was the element that received the lowest rating (6.52 out of 10).

During the online sessions, the chat was always enabled for sharing impressions or doubts, for technical assistance requests to exchange opinions, and for questions among peers and with the facilitators. In the practical sessions, the group was often divided into smaller teams to discuss specific cases or do practical activities.

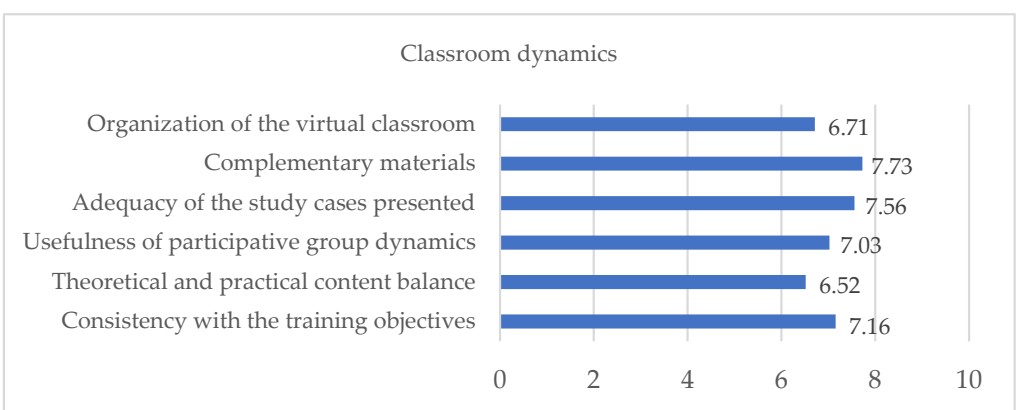

**Figure 2.** Classroom dynamics.

The care towards the participants was evident in the flexibility for submitting homework assignments and in the solutions provided by the facilitators when some participants reported difficulties in completing the activities. While the observed sessions adhered to the starting and ending times, one of the notes made in the questionnaires referred to the intensity of the training. This described it as follows: "The facilitators and the work they do are powerful; they do a spectacular job in such an innovative field, but the training program has been very intense for me."

To evaluate the response criteria, we looked at the participants' assessment and satisfaction with classroom dynamics, as well as their perceived usefulness of the content. As shown in Figure 3, they primarily highlighted the knowledge of the facilitators/instructors (an average of 8.79 out of 10) and the training usefulness for their future careers (7.55 out of 10). On the other hand, the elements poorly scored were the possibility of retaining knowledge (5.71) and the dynamics of the online sessions (6.17). During the observed sessions, some participants mentioned revisiting the proposed strategies for password management in the days following the session. One of the participants' comments in the questionnaires that drew our attention had to do with some limitations in digital skills and

knowledge. These types of comments alert us to the digital gap that some participants experienced: "Due to the digital disadvantage I have, some of the content presented was beyond my capacity to grasp" (Participant #15). Similarly, one of the facilitators mention that, in some dynamics, it seemed that participants did not fully understand when certain technological tools were referenced. For future versions, it would be relevant to conduct an initial assessment of the participants' technological skills in order to tailor the content offered.

Concerning our last criterion—influence—in the final evaluation one participant commented on her perception of the degree of internalization of the learning: "More training would be needed to go deeper into the content and internalize the knowledge; there is a lot of content, and it is very dense" (Participant #16). This criterion of influence will be explored in greater depth in the following section, where the results of self-perceived knowledge acquisition are presented.

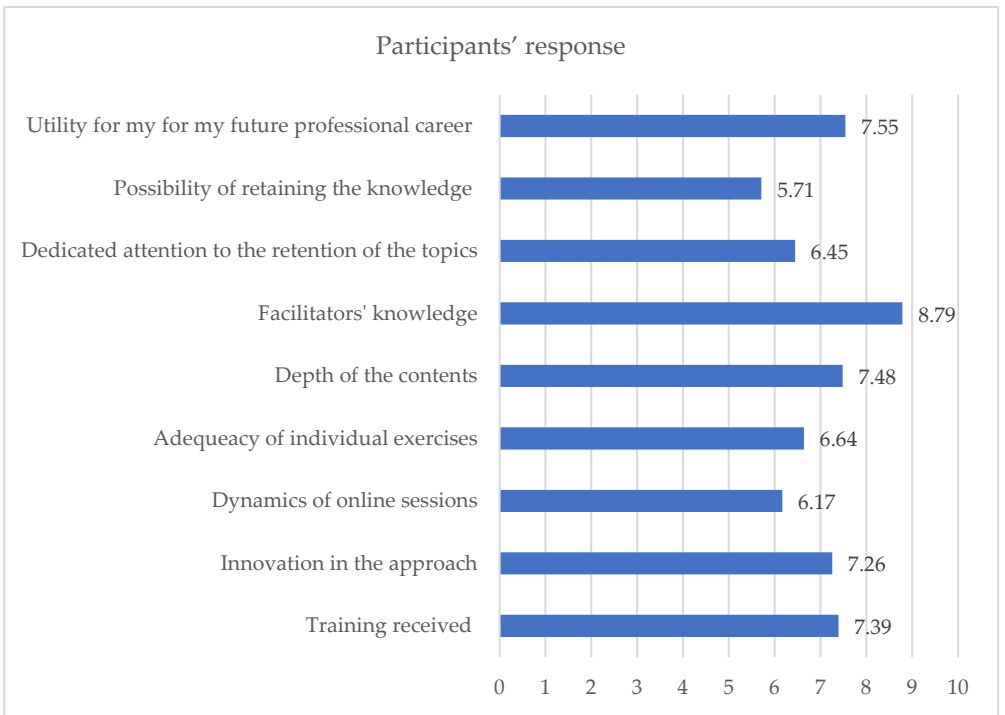

**Figure 3.** Participants' response.

*3.2. Self-Perceived Learning*

Figure 4 presents the results of the questionnaires on self-perceived knowledge acquisition about GRDV. We can see that the self-perceived learning increased by at least one point in all aspects. The knowledge areas that saw the most significant increase, indicating an enhanced perception of learning, were "I know what holistic security means", and "I know how to access different sources of information and data available on digital gender-related violence." On the contrary, the least changes were in relation to how digital gender-related violence affects both "people with public relevance", and " the health and well-being of the affected individuals." It must be noted that in both cases, participant knowledge self-perceptions of this element were already quite high before the course.

As observed in Table 3, when conducting the Student's T-test for independent samples, the *p*-value (significance) was under 0.05 for each of the twelve knowledge areas evaluated, leading us to reject the hypothesis of equal means (null hypothesis). Therefore, we can assume that there is a significant difference between the scores before (pre-test) and after (post-test) the training sessions.

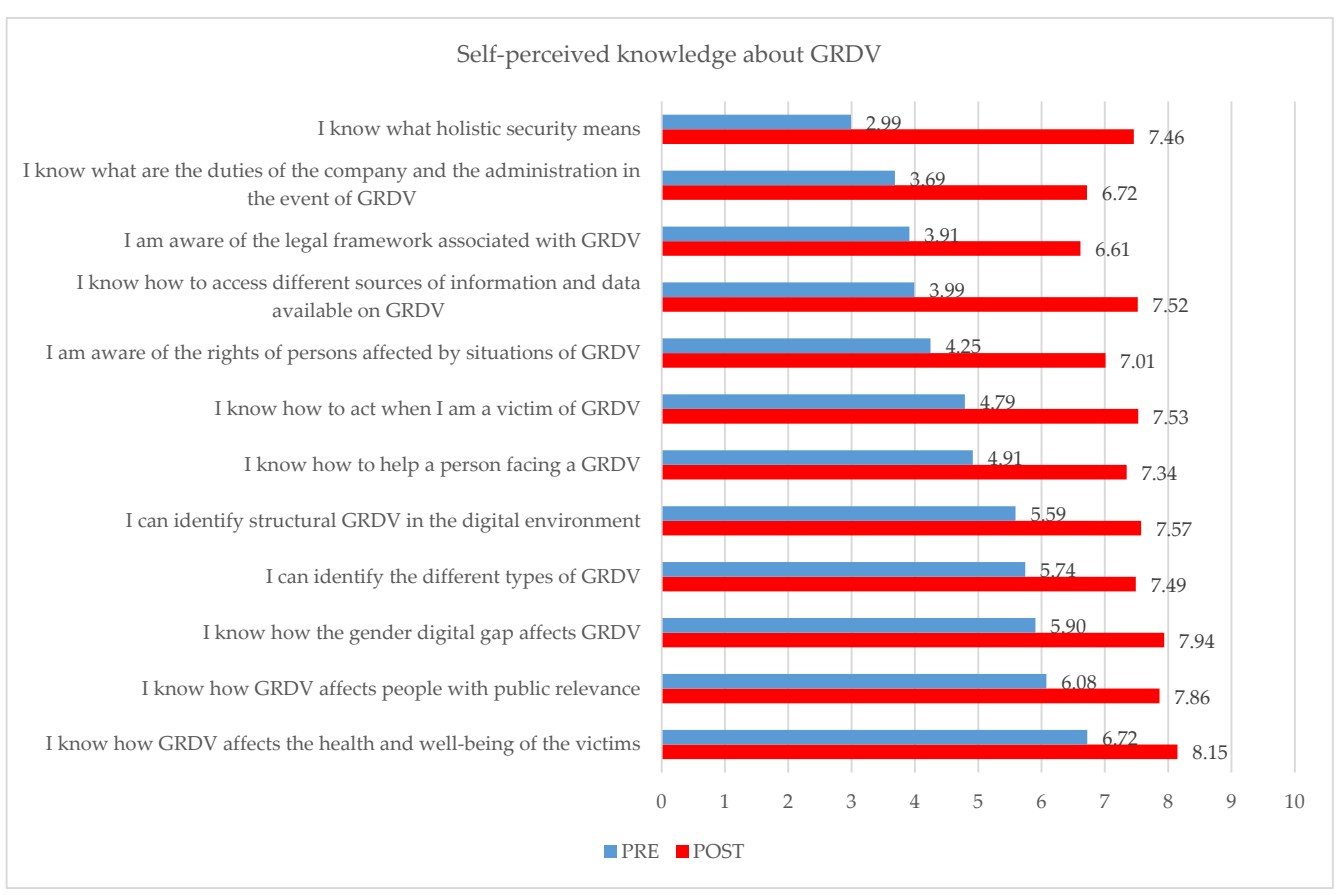

**Figure 4.** Self-perceived knowledge about GRDV.

**Table 3.** Student's T-test for independent samples—*Self-Perceived Knowledge about GRDV*.

| | | Levene's Test for Equal Variances | | T-Test for Equality of Means | | | | | | |
|---|---|---|---|---|---|---|---|---|---|---|
| | | F | Sig. | t | df | Sig. (2-Tailed) | Mean Difference | Std. Error Differ. | 95% Confidence Interv. Difference | |
| | | | | | | | | | Inferior | Superior |
| IDENTIFY GRDV | Equal variances assumed | 8637 | 0.004 | −7945 | 199 | 0.000 | −1830 | 0.230 | −2284 | −1376 |
| | Equal variances not assumed | | | −8088 | 181,063 | 0.000 | −1830 | 0.226 | −2277 | −1384 |
| IDENTIFY STRUCTURAL GRDV | E.V.A. * | 5548 | 0.019 | −7711 | 199 | 0.000 | −1899 | 0.246 | −2385 | −1413 |
| | E.V.N.A. ** | | | −7810 | 191,466 | 0.000 | −1899 | 0.243 | −2379 | −1419 |
| AFFECTS GRDV | E.V.A. | 8574 | 0.004 | −5420 | 199 | 0.000 | −1422 | 0.262 | −1939 | −0.905 |
| | E.V.N.A. | | | −5513 | 182,981 | 0.000 | −1422 | 0.258 | −1931 | −0.913 |
| AFFECTS GRDV PR | E.V.A. | 11,930 | 0.001 | −7148 | 199 | 0.000 | −1788 | 0.250 | −2282 | −1295 |
| | E.V.N.A. | | | −7268 | 184,133 | 0.000 | −1788 | 0.246 | −2274 | −1303 |
| GENDER GAP | E.V.A. | 7304 | 0.007 | −7502 | 199 | 0.000 | −2033 | 0.271 | −2567 | −1498 |
| | E.V.N.A. | | | −7614 | 187,954 | 0.000 | −2033 | 0.267 | −2559 | −1506 |
| LEGAL FRAMEWORK | E.V.A. | 9658 | 0.002 | −9047 | 199 | 0.000 | −2700 | 0.298 | −3289 | −2112 |
| | E.V.N.A. | | | −9165 | 191,326 | 0.000 | −2700 | 0.295 | −3281 | −2119 |
| SURVIVOR'S RIGHTS | E.V.A. | 16,411 | 0.000 | −10,101 | 199 | 0.000 | −2763 | 0.274 | −3302 | −2223 |
| | E.V.N.A. | | | −10,286 | 180,534 | 0.000 | −2763 | 0.269 | −3293 | −2233 |
| COMPANY DUTIES | E.V.A. | 13,942 | 0.000 | −10,633 | 199 | 0.000 | −3033 | 0.285 | −3596 | −2471 |
| | E.V.N.A. | | | −10,785 | 188,988 | 0.000 | −3033 | 0.281 | −3588 | −2478 |
| HOLISTIC SECURITY | E.V.A. | 17,664 | 0.000 | −14,399 | 199 | 0.000 | −4468 | 0.310 | −5080 | −3856 |
| | E.V.N.A. | | | −14,588 | 191,154 | 0.000 | −4468 | 0.306 | −5072 | −3864 |
| SOURCES INFORMATION | E.V.A. | 14,277 | 0.000 | −11,896 | 199 | 0.000 | −3530 | 0.297 | −4116 | −2945 |
| | E.V.N.A. | | | −12,106 | 182,026 | 0.000 | −3530 | 0.292 | −4106 | −2955 |
| HELP | E.V.A. | 11,296 | 0.001 | −8983 | 199 | 0.000 | −2429 | 0.270 | −2963 | −1896 |
| | E.V.N.A. | | | −9137 | 183,330 | 0.000 | −2429 | 0.266 | −2954 | −1905 |
| ACT | E.V.A. | 12,035 | 0.001 | −9594 | 198 | 0.000 | −2736 | 0.285 | −3298 | −2173 |
| | E.V.N.A. | | | −9810 | 176,180 | 0.000 | −2736 | 0.279 | −3286 | −2185 |

Note: * E.V.A.: Equal variances assumed. ** E.V.N.A.: Equal variances not assumed.

### 3.3. Skills Improvement (Evaluation Team Perspective)

Figure 5 presents the results of the evaluation of the fictional case. It shows an increase in both diagnosis—identifying the GRDV presented in the case—and prescription—indicating response strategies to GRDV. The average score for diagnosis increased by 1.85 points, while the average score for prescription increased by 1.26 points.

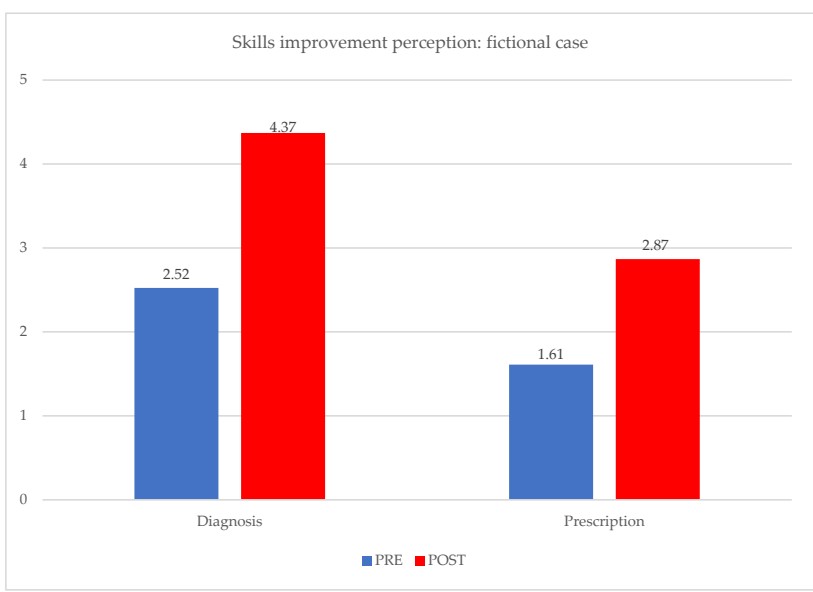

**Figure 5.** Skills improvement perception—fictional case.

Table 4 presents the results of the paired samples T-test and allows us to confirm that the evaluators' perception of skills improvement among the participants was significant (with a *p*-value below 0.05), both in diagnosis and prescription. Therefore, we can reject the null hypothesis and assume that there is a significant difference between the scores before and after the training sessions regarding the perception of skills improvement from a third-party perspective.

**Table 4.** Paired samples T-test—skills improvement (evaluators' team perception).

| Pairs | | Paired Differences | | | | | t | df | Sig. (2-Tailed) |
|---|---|---|---|---|---|---|---|---|---|
| | | Mean | Std. Deviation | Std. Error Mean | 95% Confidence Interval of the Difference | | | | |
| | | | | | Inferior | Superior | | | |
| Pair 1 | Diagnosis PRE & POST | −1831 | 0.834 | 0.109 | −2048 | −1613 | −16,867 | 58 | 0.000 |
| Pair 2 | Prescription PRE & POST | −1271 | 1284 | 0.167 | −1606 | −0.937 | −7603 | 58 | 0.000 |

## 4. Discussion and Conclusions

The analysis implemented proved that the training sessions were beneficial for generating greater learning outcomes (Raab and Stuppert 2018). In fact, it showed that self-perceived knowledge increased for each assessed area and that there was a clear improvement in training participants' perceived skills, both in GRDV diagnosis and prescription. The implemented evaluation also allowed us to detect the difference between improvement of self-perceived knowledge and increased skills from the evaluator's standpoint, which is coherent with the fact that online training is a good option for training practical skills to address gender-related violence (Etherington et al. 2017). The short duration of the interventions may explain why there is no greater growth in perceived skills (Pagani et al. 2022). In fact, Sharma et al. (2022) emphasize that training programs for professionals

working with survivors of gender-related violence should be viewed as an ongoing process rather than a one-time event. Continuous follow-up with participants over time could also provide insights into the extent of understanding and application of the training contents by the participants.

Moreover, the experience we presented here also allows us to reflect on the questions we posed ourselves at the beginning of the article concerning the possibility and the pitfalls of improving GRDV-training evaluation schemes from a feminist perspective. In the first instance, our results reveal the importance of incorporating different evaluation strategies, both qualitative and quantitative ones, as well as the value of reflexivities in feminist research practices (Jenkins et al. 2019). As suggested by Azpiazu and Luxán Serrano (2023, p. 4), "the development of feminist methodological instruments is closely related to the material experience of people and their presence in the spaces of science production. Reflecting in a circular and shared way on knowledge processes is the only way in which an adequate corpus can be built so that those who need it can use it and re-evaluate it critically and reflexively."

Therefore, the exercise of this article aims to share scientific construction strategies outside the circuits of capital recovery. In this sense, we hope that this text and the experience on which it is based, with its strengths and its limits, can also serve to support the next gen(d)eration of scholars, such as Author1, to introduce critical feminist methodologies in their own work (Wigginton and Lafrance 2019).

In fact, we have shown that the integration of several of the feminist assessment criteria developed by Biglia et al. (2022) contributes to the production of more robust evaluations thanks to the integration of methods and perspectives (the participants, the researchers, and the experts). This is needed in a context where gender politics and policy seem to be mostly implemented as a response to a mandatory norm instead of being based on a real commitment to social transformation. It is at this moment where the evaluation of GRV(D) training needs to be accurate, and feminist, to detect when and how the implemented practices promote real changes instead of being mere patches of the cisheteropatriarcal governance process (Jiménez 2022).

An evaluative approach such as the proposed one is transformative in itself, "allow[ing] for pluralism, creat[ing] scope to highlight differences and, enabl[ing] the contestation of interests, views, and knowledge claims. [It helps] to better understand the mechanisms that do accomplish successful empowerment in co-production projects in isolated projects and the ways that those mechanisms can be connected to, or embedded within, broader processes of societal transformations." (Turnhout et al. 2020, p. 43).

Nonetheless, our experience also evidences the difficulty of fully implementing such a strategy. This is so, in the first place, because many hours of work are required for the collection, registration, and analysis of information; and long-term observation and analysis will also be required, especially for the diffractive criteria. These needs contrast with the productivist logic of neoliberal research centers and policymakers that frequently fail to understand and, therefore, invest in the evaluations of gender equality programs (Kalpazidou Schmidt et al. 2023). Additionally, the specificity of gender-related violence response services asked for an adaptation of the training process to the involved professionals, which makes it more difficult to standardize a common evaluative process. We must remember that training configured as a socially sensitive intervention cannot be homogenously evaluated because it is necessary to capture the "ecology of relationships" that impact each program implementation and its outcomes (Cahill et al. 2019).

In this sense, the evaluative feminist proposal (Biglia et al. 2022) must be intended as an open tool to be adapted with care by researchers respecting the specificities of each situated context, as we hope to have been able to show in our text.

**Author Contributions:** C.G.-S.: writing—original draft, resources, investigation. J.B.-M.: conceptualization, methodology, investigation, data curation, formal analysis, validation, writing, funding acquisition and supervision. B.B.: conceptualization, methodology, validation, writing—review and editing. All authors have read and agreed to the published version of the manuscript.

**Funding:** eGBVHelp (101005742) Tackling and responding to online gender-based violence through a pioneering e-helpline for reporting GBV online and empowering women, girls and LGBTIQ+ persons and professionals, Funding by European Commission (REC-RDAP-GBV-AG-2020). The contents of this publication do not necessarily reflect the position or opinion of the European Commission.

**Institutional Review Board Statement:** Ethical review and approval were waived for this study due to the research project "Tackling and responding to online gender-based violence through a pioneering helpline for reporting GBV online and empowering women, girls and LGBTIQ+ persons and professionals (eGBVHelp!)" does not use any experimental procedures or interaction with underage participants. The pre- and post-questionnaires used to collect information on participants' knowledge and skills did not involve sensitive information and therefore did not require the approval of an ethics committee by the University of Barcelona or the European Commission. The research followed all the procedures required by the Code of Conduct for Research Integrity of the University of Barcelona.

**Informed Consent Statement:** Informed consent was obtained from all subjects involved in the study.

**Data Availability Statement:** The datasets used during the current study are available from the corresponding author upon reasonable request.

**Conflicts of Interest:** The authors declare no conflict of interest.

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
