# Peer review of "Evaluation of Gender-Related Digital Violence Training in Catalonia"

_socsci, doi:10.3390/socsci13020096_

Round 1

Reviewer 1 Report

Comments and Suggestions for Authors

Your introduction could offer a fuller discussion of the expansion of digitisation, to give greater context to the relevance and value of your research. The current discussion is very limited and refers to ‘digitisation’ in a very general sense, without more specific reference to the ways in which digital technologies and digital environments are shaping social relations, inequalities and identities, within a specific context. For example, you mention that your training was delivered to healthcare professionals, so some context in relation to the healthcare sector would add strength.

At the point at which you introduce the concept of gender-related violence it would be helpful to offer some background to the use of this term, as opposed to the term gender-based violence. For example: BIGLIA, B. and ALLDRED, P., 2015. Gender-related violence and young people: an overview of Italian, Irish, Spanish, UK and EU legislation. Children and Society, 29 (6), pp. 662-675. ISSN 0951-0605

In section 1.1 you could have contained your discussion of the search strategy employed within your literature review in favour of a more developed discussion of your conceptual framework.

The tables used throughout are clear and add clear value in representing both aspects of the data collection approach, and insights from your analysis of the data collected.

You set out very clearly how the data presented addresses the research objectives.

I liked your discussion of participant motivation, as it provided helpful context in considering participant responses in relation to training outcomes. Although this aspect was not drawn out in your discussion as fully as it could have been.

In your discussion I thought there was a missed opportunity to reflect on participant expectations of the training, and specifically that so few participants held an expectation of engaging in discussions with the rest of the group. This seemed to present a challenge in relation to aspects of your approach to training delivery.

The start of your discussion and conclusions section suggests ‘This last section summarizes the results of this work and puts them in dialogue with the literature consulted.’ Yet there was very limited dialogue with the literature and the sub-section titled: ‘The digital gap and the need for an empowering training’ seemed to be presenting new observational data, rather than developing your discussion of the data already presented.

Your conclusions were not stated as explicitly as they might have been. I did think there were conclusions present in the discussion and conclusions section, but they could have been more explicitly stated at the end of this section.

Comments on the Quality of English Language

General: The term ‘trainings’ tends to be used through, where ‘training’ would be more appropriate

Line 40: ‘result of training provided…’, instead of ‘results of a training provided…’

Line 43: ‘action-research’ as opposed to ‘research-action’

Line 52: should this read ‘empowerment-oriented’ as opposed to ‘empowering-oriented’

Line 103-104: You start this sentence with ‘In the words of 103 Nancy Fraser…’ but you not then include a quote, suggesting you are not using the words of Nancy Fraser. Perhaps worth going to Fraser’s original work?

Line 121: remove ‘more’ from this sentence: ‘without incorporating more other elements…’

Line 140: is there a more appropriate term you could use than personnel? For example, professionals or health care professionals?

Line 172: ‘Regarding to the methodological design’ should read ‘Regarding the methodological design’

Line 426: should this read ‘it is particularly noteworthy…’?

Figure 4 presents a very clear representation of the pre- and post-training responses, I don’t think that Table 3 is also needed.

Line 466: would be better labelled as ‘Discussion and Conclusion’ on the basis that this is the order in which they are addressed.

Author Response

Dear reviewer,

We greatly appreciate your thoughtful comments that helped to improve the quality of our manuscript. Firstly, we are aware that our first version did not meet the level of English required to publish in Social Sciences; because of this, we have had the whole manuscript revised by an English native editor in order to improve the quality of writing and we have introduced your comments (7-15) about the quality of the English language. Besides, we have reordered its structure and rewritten its different sections in order to be more clearly understandable. We think that the reading of the manuscript is clearer now.

On the other hand, following your recommendations, (1) we have changed the focus of our introduction from digitalization to the evaluation of training in gender related-violence, as other reviewers suggested, as we consider that it is more appropriate to the aim of the article. Also, we have more clearly specified the structure of the training program and who its recipients were, as you suggested. We think that it is clearer now. (2) The article is more focused now on GRV than it was before, as you suggested, thanks to the addition of some references and the discussion of that subject. (3) We have changed the literature review section, too, in order for it to be more connected with our purpose; in relation to (4) and (5), we have rewritten the whole discussion section so that it is better related to the objectives of the article. Thank you very much for your kind recommendations.

Kind regards,

Reviewer 2 Report

Comments and Suggestions for Authors

The development of evaluations of training processes is a subject that requires study. And, in that field, this work is especially relevant when working on gender-related digital violence.

However, in view of its publication, we believe that a thorough review is necessary. The following comments are intended to aid in that improvement.

In general terms, we can say that the main problem is that the importance of the scope of the results is not clear and, also, a certain disconnection is detected in the development of the discourse. It would also be interesting if authors could explain the originality of the text more explicitly. 

DISCONNECTION

Conceptual framework: 

- If this is not a prospective theoretical study, why is a systematic literary review carried out? The authors need to add a guide and an article to the results – and the reference to the same text (Simões et al.) appears repeatedly-, which suggests that perhaps it would have been more logical to create a theoretical framework based on the bibliography that responds to the authors' point of view (feminist perspective).

- Taking into account the objectives of the text, why it is necessary to collect the different references of the phenomenon analyzed? Authors use the notion “gender-related digital violence” from the beginning. Hence, it is not necessary the list of references. 

Methodology

- Is it necessary to talk about feminist evaluation in the theoretical framework? I think it would be better in methodology, because it's the basis of empirical work.

- It is not clear why authors talk about “feminist evaluation”: The indicators could be applied to any training process. It does not measure, for example, the presence of a feminist perspective in a project or the ethical values that are worked on.

- 347-349: “Previous studies have demonstrated the feasibility of conducting remote evaluations of gender-related violence interventions”. If this idea is not related to training (instead, interventions), perhaps it is not necessary to explain it. 

- Is it necessary to talk about “the type of users served by the institutions

where the participants (works)”(271)?

- “Motivations of the participants” (370) is related to the sample. Hence, it is part of the methodology explanations. That information is not part of the results: It does not appear among objectives. 

THE SCOPE OF THE RESULTS

- The text offers a lot of details and does not allow us to see what is really important. And sometimes it leads to contradictions, for ex. “Online a viable option … to capacitate.. ” (126) - “online training may not be the best option for training” (488) / School (137)- School and universities (153) 

- It is necessary more information about FemBloc (Budget, projects, history, team,…). Explaining the relevance of this project can help to understand the importance of the article. That information may appear in introduction.

- Objectives are rather microobjectives. In order to analyze the results, it would be necessary to have more information about the development of the training. That would make it possible, for example, to establish some relationship between how each aspect was treated and the results. As the study develops, the results do not allow us to know which training strategies worked and which did not. We cannot interpret the results if we do not know the dynamics applied.

- Sample: 104 people. Experience, gender identity, motivation of participation or institution (for example) can be used as an analysis variable. This would help to better exploit the results

Conclusions: The conclusions must not be a summary. To improve that, authors can talk, for example, about the effectiveness of the applied evaluation. 

Limitations explained are specific issues of fieldwork. Instead, limitations have to be defined on the basis of the limitation of the scope of the results. 

Other comments

-(223-229) If authors improve the writing, this paragraph is not necessary

-(302-304) “the potential risk of experimental mortality effect (Tejedor, 1981) may exist because it is likely that participants who dropped out of the training were also those with less interest and/or knowledge about GRDV” I do not understand that sentence in that article. Authors talk about the evaluation of am especific training experience, so it is not necessary to think about that. It is more questionable that a control group has not been applied, as authors say. 

-Sample citations are not identified (at least, authors can give the number of the participant? But it would be better if the add qualitative information as gender or age, for example).

-Links to questionnaires do not work (table 2) 

-Table 4 is not clear.  It's hard to identify the lines in the table

Author Response

Dear reviewer,

We greatly appreciate your thoughtful comments that helped improve the manuscript. We have had the manuscript revised by an English native editor in order to improve the quality of the English language, reordered the structure of the manuscript, and rewritten the different sections in order for it to be more clearly understood.

As you suggested in (1), we rewrote the abstract, the introduction, and the discussion section to better explain the originality of the text in relation to previous studies, with respect to the improvement of the evaluation of training in gender-related violence. In relation to (2), we have changed the focus of our literature review to the evaluation of training in GRV, and we have rewritten the objectives (3) to better fit the purpose of the article. Besides, we added some references to GRV at the beginning of the article (4). We think that the introduction and the conclusion are now better connected to each other than they were before. As you suggested in (5) and (6), we have moved the references to feminist evaluation to the methodology section and rewritten it according to the reviewer’s suggestions. On the other hand, we consider that the English language revision contributes to an improved understand of this section. As you suggested, we have removed these sentences (8) and rewritten this part of the section, reordering the motivations of the participants (9) and being more explicit about the results (10). In relation to the Fembloc project (11), the whole information will be in the project section of the article when the manuscript is accepted for publication following the instructions of the journal. We have rewritten the objectives (12) and we have added more information about the development of the training program, as you suggested. (13) We have modified the writing of the objectives to make them more understandable, as it was not necessary to analyze some variables (14). Finally, we have rewritten the discussion and conclusion sections so that our conclusions are more explicitly stated.

Kind regards,

Reviewer 3 Report

Comments and Suggestions for Authors

Dear authors

I have important concerns about your manuscript. This work is really hard to follow. The quality of the writing is low, and it would be a good idea to hire an English native editor to catch some expressions/sentences that could be presented more clearly. Second, the structure of your manuscript must be improved. Please, consider the organizational pattern used in academic research: introduction, methods, results, discussion, and conclusion. Also, be careful in including only the information that each section requires. My perception of your work is that must be carefully organized in order to make it readible. In specific, make sure the information graphically presented is accurated in its form and content. Fourth, it is hard to understand if your instruments are scales, interviews, or something else. In the case you are using scales, you should add information in your method section about it (you should report the psychometric properties as well). 

Comments on the Quality of English Language

This work is really hard to follow. The quality of the writing is low, and it would be a good idea to hire an English native editor to catch some expressions/sentences that could be presented more clearly. 

Author Response

Dear reviewer,

We greatly appreciate your thoughtful comments that helped improve the manuscript. We have had the manuscript revised by an English native editor, as you suggested, in order to improve the quality of the English language; we have reordered the structure of the manuscript and rewritten the different sections to make them more readily understandable. We have explained the different instruments used in the evaluation (pre-test questionnaire, post-test questionnaire, two fictional cases, and an observation diary). We no longer include scales in the questionnaires, whereby we do not report psychometric properties.

Kind regards,

Round 2

Reviewer 2 Report

Comments and Suggestions for Authors

Text has improve after revision. But I think that the last point, Discussion and conclusions, can improve. Now it is an abstract of the results, a good abstract, but I think authors can give deep information about the evaluation proces, its importance and usefulness . Authors can take advantage of this section to explain global conclusions and no repeat what they have said on results section.

Figure 1. Motivations and Expectations of the Participants. Repetition: Figure 1. appears in "2.1. Population and Saple" and in "3. Results". Authors have to decide where it must be explaned. It seems that is better in "Results"

Lines 302-303: "based on the criteria of (1) positionality, (2) classroom interaction, (3) care, (4) response, and (5) influence". Is adding number necessary? 

Errata need to be checked. I mention that ones I have detected:

Line 50: "The existing). In this line, Simões et al. (XXX) literature" 

Line 57: "(Etherington et al., 2017). According to the literature"

Lines 131-133: Repetition /  "To close this section, we could highlight three key ideas. Firstly, the lack of rigorous evaluations despite the increasing number of gender-related violence training programs. To close this section, we could highlight three key ideas. Firstly, the lack of rigorous eval-"

Lines 146-147: Repetition / "2. Methodology 2. Methodology"

Line 285: "Table 2. Data collection tools" / Missing Pre Questionnaire

Line 391: "rience,. Taking"

Author Response

Thank you very much for your care and useful suggestions.

To answer your query, we have:

  1. Make minor adjustments to the text to improve its readability (marked with track changes); in this sense, we also propose a minimum change in the article title.
  2. Moved the information about participant expectation and motivation and Figure 1 in the results sections.
  3. Subsumed the reduced version of the population and sample section with the Teams description (now in subsection 2 Population, sample, and the implementation and evaluation teams)
  4. Move some of the information of the conclusion in line 320-328 (marked in blue)
  5. Rewrote the discussion and conclusion section to introduce a wider feminist methodological debate as suggested; for doing so, we have also included seven new bibliographical references, listed in blue in the bibliography.

We hope to have been able to satisfactorily address all your suggestions, and we thank you for helping us to make the article stronger.